⊖ | **Open Peer Review** | Host-Microbial Interactions | Research Article

# Anti-bacterial activity of dermcidin in human platelets: suppression of methicillin-resistant *Staphylococcus aureus* growth

Erxiong Liu,[1] Shunli Gu,[1] Weizhen Xi,[2] Wenting Wang,[1] Jinmei Xu,[1] Ning An,[1] Lingling Zhang,[1] Jiajia Xin,[1] Xingbin Hu,[1] Yaozhen Chen,[1] Qunxing An,[1] Wen Yin[1]

**ABSTRACT**  Methicillin-resistant *Staphylococcus aureus* (MRSA) is one of the most common drug-resistant bacteria that cause community and hospital infections. As one of the most common "superbugs" and the pathogen with the highest global incidence of hospital-acquired infections, MRSA has developed resistance to multiple antibiotics, posing a serious threat to public health. The rapid emergence and spread of multidrug resistance have increased the urgent need for new antimicrobial strategies and agents to combat MRSA-associated infections. In recent years, platelets have been widely recognized to play an important role in human immune defense. We have previously reported that platelets inhibit MRSA by inducing hydroxyl radical (OH$^\bullet$)-mediated apoptosis-like cell death. To further explore the platelet antibacterial mechanism, supernatants from co-culture of platelets and MRSA *in vitro* were used for proteomic analysis. Based on our observations using confocal and immunoelectron microscopy, we found a previously unrecognized platelet antimicrobial peptide, dermcidin (DCD), in the alpha (*a*-) granules. Furthermore, after co-culturing with MRSA *in vitro*, activated platelets secreted large amounts of DCD. Additionally, we confirmed that DCD displayed anti-MRSA activity in a concentration-dependent manner and contributed to the inhibition of MRSA growth by platelets *in vitro*. Our findings provide important insights into the immune defense functions of platelets.

**IMPORTANCE**  The emerging multidrug resistance in many pathogenic bacterial species poses a serious problem worldwide. Methicillin-resistant *Staphylococcus aureus* (MRSA), one of the most common gram-positive pathogenic bacteria, has also evolved strains with multidrug resistance. This calls for the urgent development of novel and effective treatments or bactericidal agents to mitigate this issue in clinical settings. In this study, we identified for the first time a previously unrecognized platelet antimicrobial peptide, DCD, which impedes the proliferation of MRSA and promotes the antibacterial effect of platelets on MRSA. Our findings enrich our understanding of platelet physiological function and antibacterial mechanisms and provide new insights into the development of novel natural antimicrobial agents for controlling infections.

**KEYWORDS**  MRSA, platelet, antimicrobial peptide, dermcidin, suppression

**Peer Reviewer** Diptaraj Chaudhari, Wayne State University, Detroit, Michigan, USA

Address correspondence to Yaozhen Chen, zhenzhenscu@126.com, Qunxing An, bestar01@163.com, or Wen Yin, yinwen@fmmu.edu.cn.

Erxiong Liu, Shunli Gu, and Weizhen Xi contributed equally to this article. Author order was determined by the corresponding authors after negotiation.

The authors declare no conflict of interest.

See the funding table on p. 14.

The widespread use of antibiotics has led to the emergence of multidrug-resistant bacteria, which now pose a serious threat to public health (1–3). Methicillin-resistant *Staphylococcus aureus* (MRSA) is one of the most common multidrug-resistant strains causing hospital and community infections (4, 5). MRSA invasion can not only lead to localized skin and soft tissue suppurative infections in humans but also cause fatal invasive infections such as endocarditis, osteomyelitis, and pneumonia (6, 7). However,

in addition to methicillin, MRSA is resistant to $\beta$-lactams, erythromycin, clindamycin, tetracycline, norfloxacin, and other antibiotics (2, 8, 9). Unfortunately, clinical studies have shown that the effectiveness of vancomycin against MRSA is decreasing, and multi-drug-resistant "superbugs" such as vancomycin-resistant *Staphylococcus aureus* have emerged (2, 10–12). Therefore, there is an urgent need to develop new antibacterial agents and strategies to control MRSA infections.

Platelets are anucleated cells produced by the degranulation of megakaryocytes in the bone marrow (13, 14). Recent studies have shown that platelets play an important role in inflammation and host immune defense, in addition to their classical functions of coagulation and hemostasis (15–20). In peripheral blood, platelets have a life cycle of approximately 7–10 days and serve as "sentinels" against tissue damage and microbial threats (21–23). They constantly monitor vascular integrity and closely coordinate the transport and function of many cell types, helping the body build an effective immune system to fight infections and cancers (16, 24). When pathogens invade the human body, platelets can interact with them through their pathogen recognition receptors, such as Toll-like receptors, which eventually leads to platelet activation and aggregation (21, 25–28). Subsequently, the alpha ($a$-) granules of activated platelets release antimicrobial peptides (AMPs) and chemokine-derived peptides (kinocidins) (29, 30). AMPs and kinocidins, on one hand, can directly kill pathogenic bacteria (31–33); on the other hand, they can recruit and promote other immune cells such as macrophages, neutrophils, dendritic cells, and T cells to migrate and aggregate to the site of infection to exert an antibacterial effect (34–39).

We previously confirmed that platelets inhibit MRSA by inducing hydroxyl radical (OH$^{\bullet}$)-mediated apoptosis-like cell death (ALD). Platelet lysates exert the same antibacterial effect as platelets, promoting OH$^{\bullet}$ overproduction and inducing ALD in MRSA (40). Therefore, it is possible that the anti-microbial mediators present in platelets play an important role in the anti-MRSA effects. Hence, the platelet-derived antimicrobial proteins that contribute to the inhibition of MRSA growth require further investigation. In the present study, we aimed to identify the antimicrobial proteins in platelets and validate their anti-MRSA activity. At the same time, this study further improves the antibacterial mechanism of platelets and provides a new understanding of anti-MRSA research.

## MATERIALS AND METHODS

### MRSA strains, culture, and counts

MRSA (USA300, American Type Culture Collection BAA-1717, Manassas, VA, USA) was procured from the Laboratory Department of Xijing Hospital of Fourth Military Medical University. MRSA resistance has been confirmed in a previous study (40). MRSA culture and counting were performed as previously described (41). In brief, a MRSA single colony was inoculated in Luria-Bertani (LB) medium for 24 h at 37°C. MRSA was then inoculated into LB and cultured for 6–8 h until it reached the mid-exponential growth phase. Bacterial concentration was estimated spectrophotometrically at 600 nm (UV-2550; Shimadzu, Kyoto, Japan). An absorbance of 1.0 corresponded to $10^9$ colony-forming units (CFU)/mL for MRSA, according to the bacterial colony count results on plates. MRSA was diluted to $10^6$ CFU/mL in LB and set aside. Bacteria in each group were counted by serial dilution, and 100 µL was inoculated in LB plates after co-culture. Bacterial colonies were counted after incubation at 37°C for 18 to 24 h.

### Preparation of human washed platelets

Human washed platelets were prepared as described previously (40). Briefly, human apheresis platelets were centrifuged (382 × *g*, 5 min) and washed with a washing solution (phosphate buffer saline [PBS] containing 10% acid-citrate dextrose [ACD] blood preservation solution). After washing gently twice, the platelets were suspended to

achieve a final count of $150 \times 10^9$ /L in LB medium for co-culture with MRSA. The purity of platelets was determined as previously described (41). Informed consent was obtained from voluntary blood donors, and all relevant procedures were performed in accordance with the requirements of the Medical Ethics Committee of Xijing Hospital, the Fourth Military Medical University.

## Mass analysis

The coculture system of platelets and MRSA was established as follows: platelets were resuspended in LB medium to a count of $1.8 \times 10^8$/mL. MRSA was diluted to a concentration of $10^6$ CFU/mL in LB medium. Then, MRSA (600 µL) was incubated with or without platelets (5 mL) in LB medium for 10 h at 37°C with shaking at 180 rpm (the total volume of the culture system was 6 mL, in which the final concentration of platelets was $150 \times 10^9$/L, and the final concentration of MRSA was $10^5$ CFU/mL). Proteins in the supernatant of platelets cocultured with or without MRSA were enriched in cold methanol. Briefly, the supernatant sample was mixed with cold methanol (100% concentration) in a 1:5 ratio and stored at −20°C for at least 2 h. Proteins were then collected by centrifugation at 4°C ($10,000 \times g$, 15 min) and washed with methanol (90% concentration). After centrifugation at 4°C ($10,000 \times g$, 15 min), the protein precipitates were air-dried for 15 min. The final protein precipitate was dissolved in 8 M Urea protein solution (Tris-HCL [PH = 8], EDTA, DTT, urea) and stored at −80°C. Finally, the protein samples were sent to Shanghai Zhongke New Life Biotechnology Co. for quantitative proteomic analysis by tandem mass tag (TMT) technology, and the detailed experimental analysis methods are shown in File S1.

## Detection of dermcidin in platelet lysates by western blot

Platelets (approximately $10^6$) were lysed using RIPA lysis buffer (P0013C, Beyotime) for 30 min on ice. Then, platelet lysates were centrifuged at 4°C ($10,000 \times g$, 20 min), and the supernatant was collected. The protein concentration in the supernatant was determined using the BCA Protein Assay Kit (P0012S, Beyotime). Next, protein samples with a total protein content of 10, 15, and 20 µg, respectively, were separated by sodium dodecyl sulfate-polyacrylamide gel electrophoresis, transferred electrophoretically (1,200 mA/h) onto polyvinylidene difluoride membrane sheets (Immobilon-PSQ; Millipore, Germany) with a tank type blotter and then blocked for 30 min at room temperature. After incubation with a mouse monoclonal anti-DCD antibody (H-12, sc-398429, Santa Cruz Biotechnology, USA) as the primary antibody (1:1,000) that detects the C-terminus of DCD of human origin overnight at 4°C, the membrane sheets were washed with PBST (PBS containing 0.1% Tween 20) and incubated with a goat polyclonal antibody to mouse IgG (horseradish peroxidase conjugated, BA1050, BOSTER, China) as the secondary antibody (1:1,000 in blocking solution) for 1 h. After washing with PBST, the membrane sheets were analyzed using a chemiluminescence imaging system (ChemiDoc XRS+, Bio-Rad, USA).

## Detection of DCD by ELISA

A coculture system of platelets and MRSA was established as described above. After the platelets were co-cultured with or without MRSA for 10 h, the culture solution was centrifuged, and the supernatants were transferred to new tubes. The DCD concentration in the supernatant was determined using the Human Dermcidin ELISA Kit (orb406184, Biorbyt, Cambridge, UK) according to the manufacturer's instructions.

## Detection of DCD in platelets by immunofluorescence and confocal microscopy

Platelets were fixed with 200 µL of 0.05% glutaraldehyde for 15 min. After centrifuging and washing once, platelets were permeabilized with 400 µL of 0.1% Triton-X100 for 15 min at room temperature. Platelets were blocked with 3% bovine serum albumin

(BSA) in PBS for 30 min at room temperature after centrifugation. Then, the platelets were incubated with anti-DCD primary antibody (1:100) at 4°C overnight with spinning. After washing, the platelets were incubated with a goat polyclonal antibody against mouse IgG (CY3 conjugated, BA1031, BOSTER, China) as the secondary antibody (1:100) in the dark for 1 h. After washing twice, the platelets were incubated with a phalloidin working solution (PF00001, Proteintech, USA) in PBS (1:100) for 20 min at room temperature in the dark. After washing, the platelets were resuspended in 200 μL PBS, and 10 μL of them were taken on a glass slide. After covering the coverslips, the platelets were observed using fluorescence microscopy (Ni-U, Nikon, Japan) and confocal microscopy (LSM 900, Carl Zeiss, Germany), respectively.

## Localization of DCD in platelets by immunoelectron microscopy

Platelets were fixed with 200 μL of 3% glutaraldehyde for 24 h at 4°C. The fixed platelet samples were sent to the electron microscope Room of the Department of Pathology of Fourth Military Medical University for embedding and sectioning. Ultrathin sections of the platelet samples were fixed with nickel mesh, dispersed on polyester blocks, rinsed with pure water, and soaked for 5 min. After drying with a filter paper, the sections were soaked in 1% $H_2O_2$ (filtered) for 10 min. Next, the sections were rinsed, blotted dry, and incubated in a blocking solution (3% BSA in pure water) for 30 min. After blotting, the sections were incubated with anti-DCD primary antibody (1:100) at 4°C overnight. After rinsing and soaking, the sections were blotted dry and soaked in PBSA (1% BSA in PBS) for 7 min. After blotting, the sections were incubated with a secondary antibody (10 nm gold-conjugated, goat anti-mouse, ab39619, Abcam, UK) in PBSA (1:250) for 1 h at room temperature in the dark. After rinsing and drying, the sections were stained with uranyl acetate for 8 min. The sections were then stained with lead citrate for 8 min after washing, and blotting was performed. Finally, the sections were rinsed, blotted dry, and analyzed using transmission electron microscopy (TEM) (HT-7800, Hitachi, Japan).

## Antimicrobial assays of DCD-1L on MRSA

DCD-1L was synthesized by Haode Biotechnology Co. (Wuhan, China), and its amino acid sequence is as follows:

SSLLEKGLDGAKKAVGGLGKLGKDAVEDLESVGKGAVHDVKDVLDSVL.

Antimicrobial assays were performed as described previously (42). In brief, MRSA was diluted to a concentration of $10^6$ CFU/mL in PBS; 20 μL of the diluted MRSA suspension was incubated with different concentrations of DCD-1L in PBS (total volume 60 μL) for 4 h at 37°C. After incubation, the bacterial suspension was then diluted to 1:100 in PBS, and 100 μL of the diluted bacterial suspension was plated in triplicate on LB plates. Plates were incubated at 37°C for 24 h in a constant-temperature incubator, and bacterial colonies were counted. Antimicrobial activity was calculated as follows: (1 − [cell survival after peptide incubation]/[cell survival in buffer without peptide]) × 100. The $IC_{95}$ designates the lethal concentration of the current synthetic peptide in μg/mL, which leads to a 95% reduction of CFU compared with the buffer control.

## Antimicrobial assay analysis of adding DCD-1L to the co-culture system of platelets and MRSA

To further examine whether DCD-1L contributes to the inhibition of MRSA by platelets, after adding DCD-1L to the co-culture system of platelets and MRSA, we performed two experiments. First, a coculture system of platelets and MRSA was established as described above. After co-culturing for 10 h at 37°C with shaking at 180 rpm, 20 μL of the co-culture bacterial suspension was incubated with different concentrations of DCD-1L in the supernatant of the co-culture medium (total volume 60 μL) for 4 h at 37°C. After incubation, the bacterial suspension was then diluted and plated in triplicate on LB plates. Plates were incubated at 37°C for 24 h, and bacterial colonies were counted. Antimicrobial activity was calculated as described above.

Second, the co-culture system of platelets and MRSA was established as described above. DCD-1L at the final concentration of 30 µg/mL was added into the co-culture system at the same time. Then, after co-culturing for 10 h at 37°C with shaking at 180 rpm, the turbidity of the bacterial solution was observed, and the bacterial counts were counted by plate in each group.

## Statistical analysis

Statistical significance between the means of two groups was evaluated using the Student's *t* test (Prism v.5.01, GraphPad Software, La Jolla, CA, USA). The results are expressed as means ± SEM. Statistical significance was set at *P* < 0.05.

## RESULTS

### Proteomics results showed that platelet activation-related proteins were significantly changed after co-culture with MRSA

We previously confirmed that platelets inhibit MRSA by OH$^{\bullet}$-mediated apoptosis-like cell death, and platelet lysates exert the same antibacterial effect as platelets (40). Therefore, it is possible that the anti-microbial proteins present in platelets play an important role in the anti-MRSA effects. Hence, in order to clarify which antibacterial protein derived from platelets contributes to the inhibition of MRSA growth, the supernatant proteins of platelets co-cultured with or without MRSA were collected for quantitative proteomic analysis using tandem mass tag (TMT) technology. In total, 2,358 human platelet proteins were detected. Among them, 2,304 proteins with unique peptides or polypeptide segments were identified, whereas 58 proteins were not identified. According to the screening standard, multiple changes were greater than 1.4 (upregulation greater than 1.4, or downregulation less than 0.714), and the *P*-value was less than 0.05; a total of 581 differentially expressed proteins (DEPs) were screened. Compared with the PLT group, 295 proteins were significantly downregulated in the PLT-MRSA group, whereas 286 proteins were significantly upregulated (Fig. 1A and File S2). As shown in Fig. 1B, GO functional enrichment analysis of DEPs using Fisher's exact test showed that cellular process, biological regulation, regulation of biological process, and metabolic process were the main functional categories of DEPs involved in critical biological processes (BP) of platelets, among which cellular processes were enriched with the largest number of DEPs (red box labeled). The KEGG pathway enrichment analysis indicated that DEPs are located in important pathways such as platelet activation, regulation of actin cytoskeleton, ECM-receptor interaction, etc. Among these, the platelet activation pathway (red box labeled) enriched the most DEPs (Fig. 1C).

Then, we analyzed KEGG pathway maps of platelet activation. As shown in Fig. 2A, we found that multiple key DEPs are involved in platelet activation and aggregation pathways, such as von Willebrand Factor (vWF), collagen, fibrinogen, platelet glycoproteins (GPV, GPIbα), FcγRIIA, and integrin family proteins (α2β1, αIIbβ3). They are key proteins involved in the regulation of platelet activation (43, 44). Subsequently, we conducted a cluster analysis on the 31 DEPs related to platelet activation in the platelet activation pathway. The results showed that compared with the control PLT group, 31 DEPs related to platelet activation in the PLT-MRSA group were upregulated (Fig. 2B). The function of proteins *in vivo* depends on their interactions with other proteins; therefore, we constructed a protein-protein interaction (PPI) network of DEPs for platelet activation. PPI analysis revealed that vWF played a key role among the top 13 proteins involved in platelet activation (Fig. 2C). vWF has been reported to be a key factor of platelet activation (43, 45). Taken together, these results suggested that platelets may be activated after coculture with MRSA, which was closely related to some key DEPs such as vWF, αIIbβ3, collagen, and FcγRIIA.

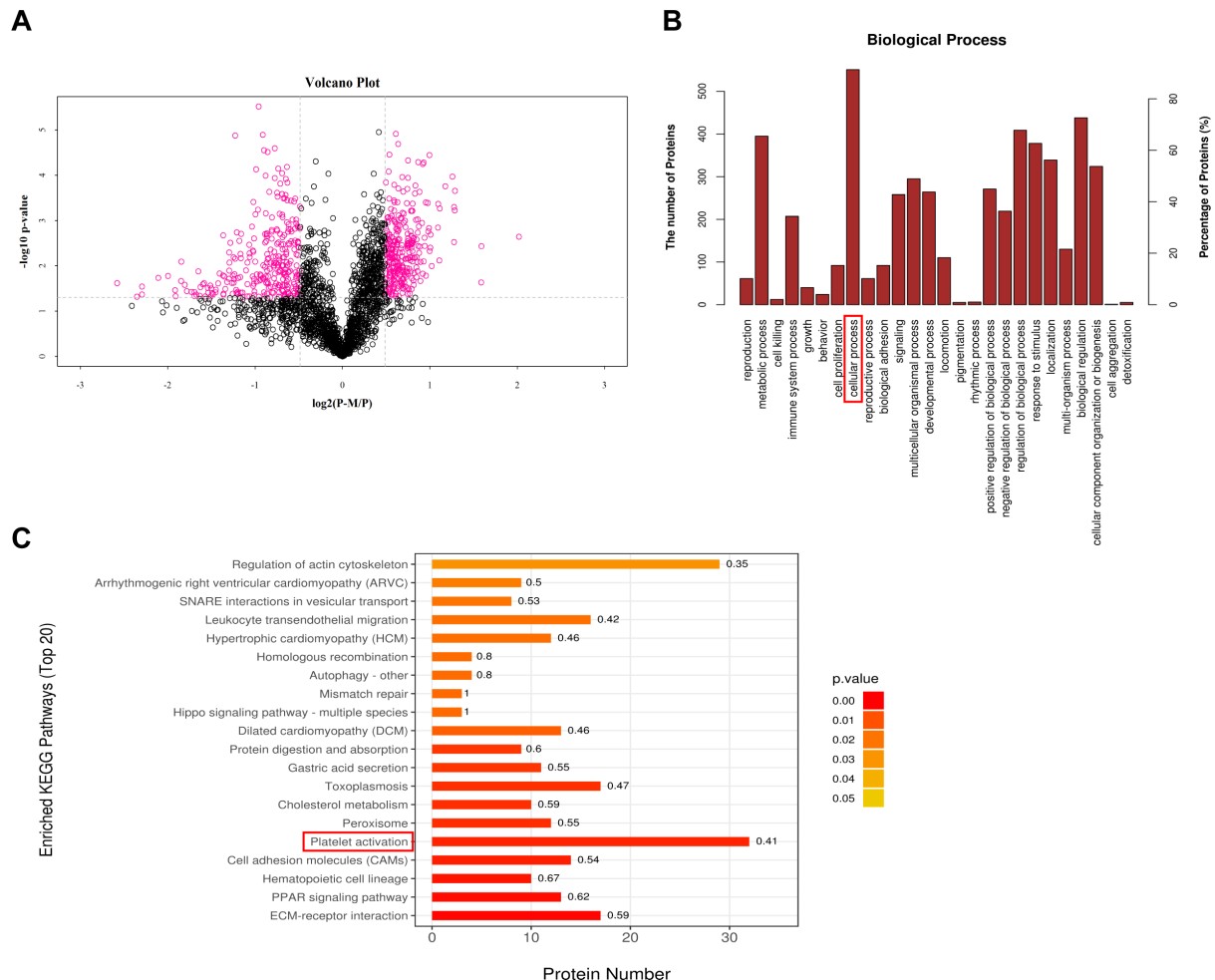

**FIG 1** After platelets co-culture with MRSA, a variety of platelet-derived proteins were significantly changed. The supernatant proteins of platelets co-cultured with or without MRSA for 10 h were collected for quantitative proteomic analysis. P, platelets culture alone; P-M, platelets co-culture with MRSA. (**A**) P-M_vs_P group volcano plots. The fold change and the *P*-value obtained by *t*-test were used to draw volcanic plots to show the significant differences between the two groups. Abscissa is the difference multiple (logarithmic transformation with base 2), ordinate is the significance of the difference, *P*-value (logarithmic transformation with base 10), red dots in the figure are the proteins with significant difference (*P* < 0.05), and black dots are the proteins with no difference. (**B**) GO functional enrichment analysis of DEPs in P-M _vs_P group. The abscissa in the graph shows enrichment to GO function classification in biological process (BP); the ordinate (left) represents the number of DEPs under each functional classification, and the ordinate (right) represents the percentage of DEPs under each functional classification in the total number of DEPs. GO, Gene Ontology. (**C**) Enriched KEGG pathways of the DEPs (Top 20) in P-M _vs_P group. The vertical Y-axis represents the pathway. The X-axis represents the number of proteins involved in the corresponding pathway. The digit at the right of each strip represents the richness factor of the corresponding pathway. Color of the bar represents the *P* value calculated using Fisher's exact test. KEGG, Kyoto encyclopedia of genes and genomes.

## Platelets release large amounts of dermcidin, a previously unknown component of platelets, after co-culture with MRSA

Based on the proteomics results, we found that DCD was significantly upregulated (File S2). DCD, which was originally isolated from human sweat, is a naturally active polypeptide (46). DCD is one of the three antimicrobial peptides (Defensins, Cathelicidins, and Dermcidin) found in humans and was first found in human sweat gland cells (47). The full-length DCD is composed of 110 amino acid residues with 19 amino acid signal peptides at the N-terminus, which are characteristic of the secreted proteins (47, 48). First, to confirm that DCD is derived from platelets, human platelets were labeled by immunofluorescence using DCD-specific monoclonal antibodies and observed by fluorescence microscopy. The results showed the presence of the DCD protein in

**A**

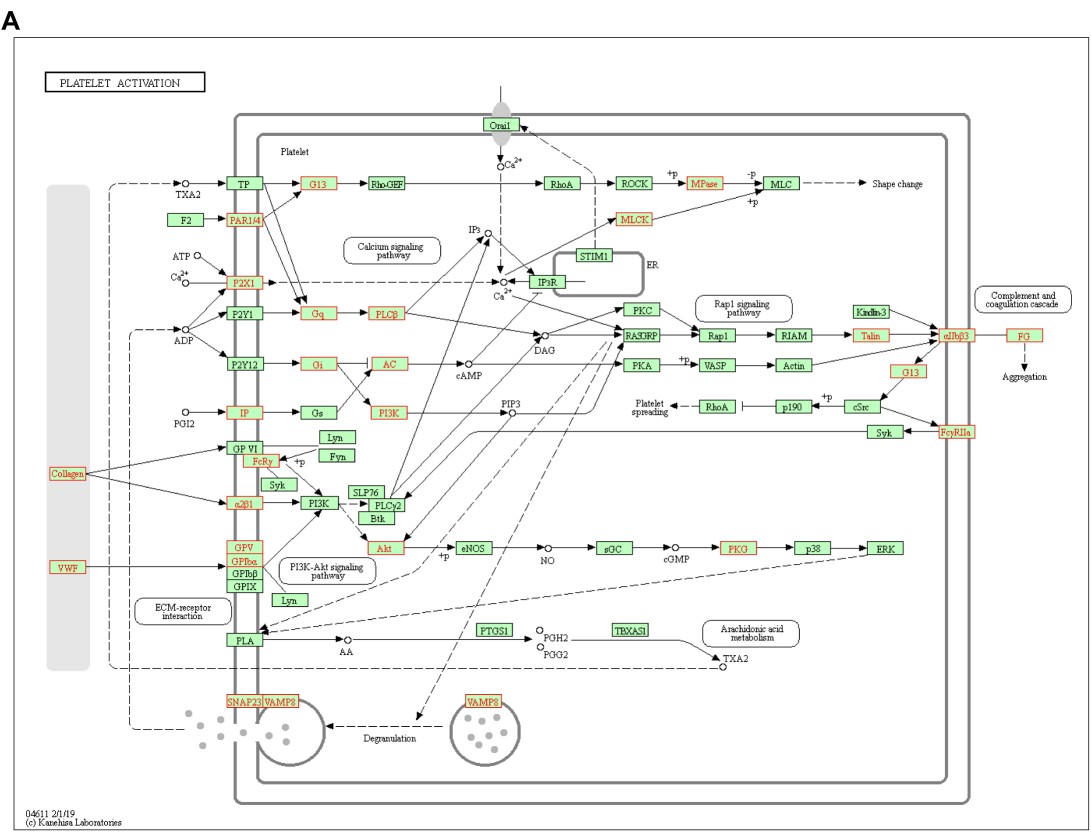

**B**

**C**

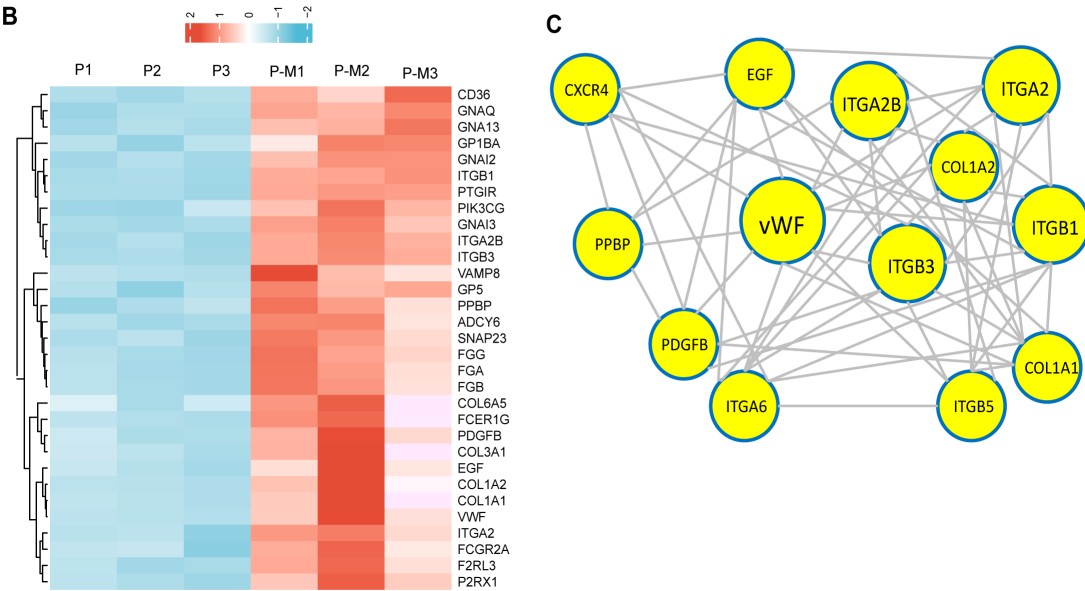

**FIG 2** Platelet activation-related proteins were significantly changed after co-culture with MRSA. (**A**) Diagram of platelet activation signaling pathway including the DEPs. The DEP proteins are shown in red. (**B**) Cluster analysis of 31 DEPs related to platelet activation between the P-M group and the *P* group. Rows represent the proteins, and columns represent the groups. The relative expression level is indicated by the intensity of the color. Red, high expression; Blue, low expression. P, Platelet; P-M, Platelet-MRSA. (**C**) PPI analysis of part DEPs on platelet activation. The DEPs are highlighted in yellow. Circles and font sizes indicate how closely the protein interacts with other proteins, and larger circles and font sizes indicate more interactions with other proteins.

platelets (Fig. 3A). Western blot analysis of platelet lysates showed that the DCD protein was present in the platelet lysates (Fig. 3B). In addition, the DCD content in the supernatant of platelets co-cultured with or without MRSA was detected using ELISA. The results

showed that compared with the platelet alone group, the secretion of DCD increased significantly after platelets co-cultured with MRSA, and the concentration reached 3.58 µg/mL (Fig. 3C). These results indicate that DCD is a previously unrecognized protein of platelets, and after co-culturing with MRSA, the platelets secrete a large amount of DCD.

## Dermcidin is present in the *α*-granules of platelets

To determine the location of DCD in platelets, we first observed the colocalization of DCD and platelets using confocal microscopy after staining human platelets with immuno-fluorescence. These results confirmed that DCD was present in the platelets (Fig. 4A). Platelets were then labeled with a specific DCD monoclonal antibody and gold-conjugated secondary antibody and observed using TEM. Immunoelectron microscopy (IEM) showed that DCD was present in the *α*-granules of platelets (Fig. 4B). These results indicated that the novel platelet antimicrobial peptide DCD, similar to other platelet antimicrobial peptides, exists in the *α*-granules of platelets.

## DCD inhibits MRSA growth in a concentration-dependent manner

As described above, we demonstrated that after co-culture with MRSA, a large amount of DCD was released from activated platelets. Full-length DCD is composed of 110 amino acid residues. DCD is then hydrolyzed and processed by a protease in *vivo* to form C-terminal peptides of 48 amino acid residues (DCD-1L), 47 amino acid residues (DCD-1), and other shorter fragments. DCD-1L and DCD-1 showed broad-spectrum antibacterial activity against a wide range of bacteria, including *Escherichia coli*, *Enterococcus faecalis*, *Staphylococcus epidermidis*, *S. aureus*, etc. (42, 46, 48). Therefore, we determined the antimicrobial activity of synthesized DCD-1L against MRSA *in vitro*. The results showed that lower concentrations of DCD-1L (5, 10, 15, and 20 µg/mL) have no antibacterial effects on MRSA. DCD-1L began to exhibit partial antibacterial activity at a concentration of 25 µg/mL, and its antibacterial activity reached more than 95% at a concentration of 30 µg/mL. When the concentration reached 35 µg/mL, DCD-1L showed 100% antibacterial activity on MRSA. Then, as the concentration of DCD-1L increased, its antibacterial activity of DCD-1L remained unchanged (Fig. 5A and B). These results indicate that the 95% inhibitory concentrations (IC$_{95}$) of DCD-1L were 30 µg/mL, and the maximum anti-MRSA concentration of DCD-1L was 35 µg/mL. Therefore, DCD-1L inhibited MRSA growth in a concentration-dependent manner *in vitro*.

## DCD promoted the inhibitory effect of platelets against MRSA

We confirmed that DCD-1L inhibited MRSA growth. To further examine whether DCD-1L contributes to the inhibition of MRSA by platelets after adding DCD-1L into the co-culture system of platelets and MRSA, we performed two experiments. First, according to the above results of the concentration of DCD secreted by platelets in the co-culture system (Fig. 3C) and the concentration of DCD-1L that began to inhibit MRSA growth (Fig. 5A and B), we added different concentrations of DCD-1L to the bacterial suspension after platelet co-culture with MRSA for 10 h. Then, the bacterial colonies were counted after DCD-1L was incubated with the bacterial suspension. The results showed that the mortality rate of MRSA was 68% when the concentration of DCD-1L added was 20 µg/mL; when the concentration of DCD-1L was 25 µg/mL, the mortality rate of MRSA was more than 95%. The mortality rate of MRSA was 100% when the concentration of DCD-1L added was 30 µg/mL (Fig. 6A and B). These results indicated that the mortality rate of MRSA increased with increasing DCD-1L concentration in the bacterial suspension after platelet co-culture with MRSA.

Second, to compare the inhibition of MRSA growth by platelets or DCD-1L alone, DCD-1L at a concentration of 30 µg/mL was added to the co-culture system of platelets and MRSA. After 10 h of co-culture, the turbidity of the bacterial suspension was compared, and the number of bacterial colonies was counted in each group. The results showed that compared with the partial antibacterial effect of platelets alone, the

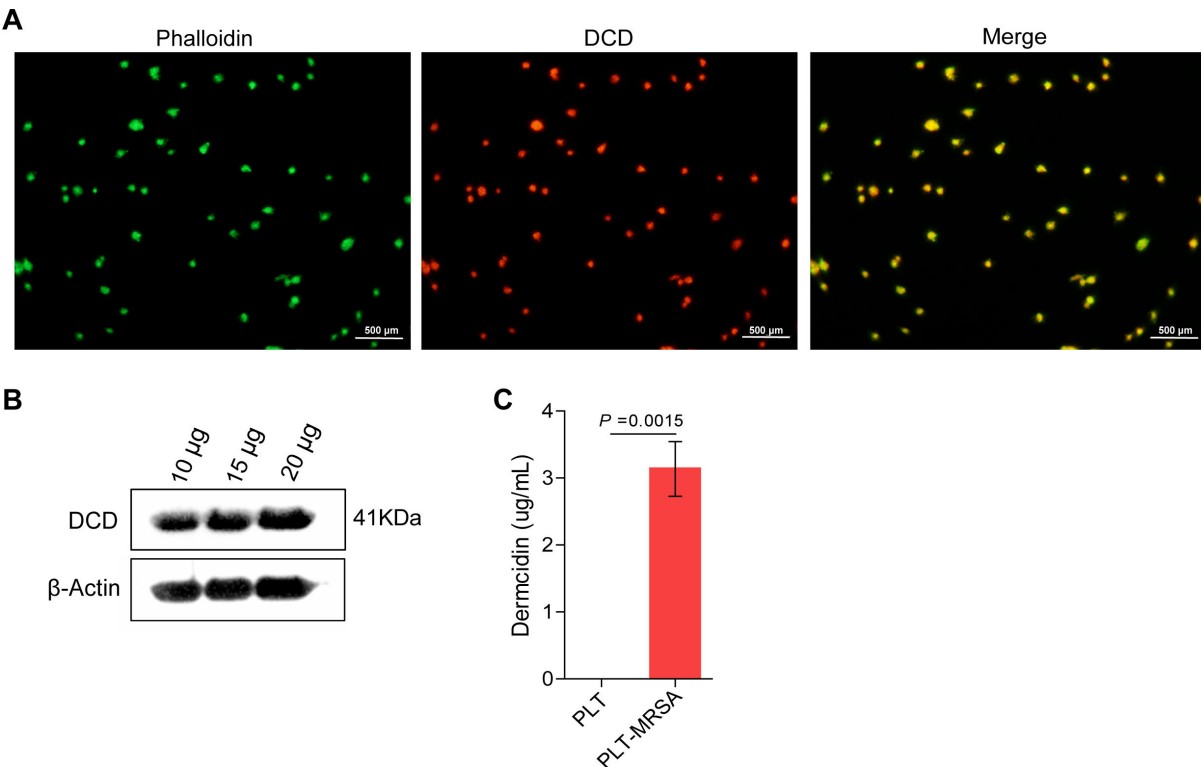

**FIG 3** Platelets secreted large amounts of DCD, a previously unknown component of platelets, after co-culture with MRSA. (**A**) Immunofluorescence detection of DCD in platelets. Phalloidin (green): CoraLite488 conjugated phalloidin, CL488-Phalloidin served as the counterstain, at dilution of 1: 100; DCD (red): Dermcidin. Scale bar = 500 μm. Images are representative of three independent experiments. (**B**) DCD expression detection in platelet lysates by western blot, 10, 15, and 20 μg, respectively, represents the added protein content of each well after platelet lysis. (**C**) Measurement of DCD content in the supernatant of platelets co-cultured with or without MRSA by ELISA. All results have been tested at least three times. Data presented as mean ± SEM. Student's *T* test for two-group comparisons.

addition of DCD-1L to the co-culture system of platelets and MRSA further enhanced the antibacterial effect of platelets, and the antibacterial effect was better than that of DCD-1L alone (Fig. 6C and D). Taken together, these data indicate that DCD-1L may promote platelet-induced inhibition of MRSA growth and that platelet-derived DCD may play an important role in the inhibition of MRSA growth by platelets.

## DISCUSSION

In this study, we identified a previously unrecognized platelet antimicrobial peptide, DCD, in the *α*-granules for the first time. After co-culturing with MRSA, activated platelets play an important role in inhibiting MRSA growth by secreting large amounts of DCD. This enriches our understanding of platelet immune function and has a profound effect on platelet function expansion. Additionally, what we found in the present study provides the theoretical and experimental basis for the research and development of new antibacterial agents and antibacterial therapy in the treatment of MRSA infection.

We found that a significant amount of DCD, stored in the *α*-granules, was released from activated platelets after co-culturing with MRSA, which contributed to the inhibition of MRSA growth. We believe that this finding is based on the following evidence. First, mass spectrometry analysis showed that platelets may be activated after co-culturing with MRSA. Second, western blotting, immunofluorescence, and IEM confirmed that DCD was derived from platelets and existed in the *α*-granules of platelets. A large amount of DCD was secreted by the activated platelets after co-culturing with MRSA, as determined by ELISA. Third, an antibacterial activity assay of DCD-1L confirmed that DCD-1L impeded the growth of MRSA in a concentration-dependent manner *in vitro*.

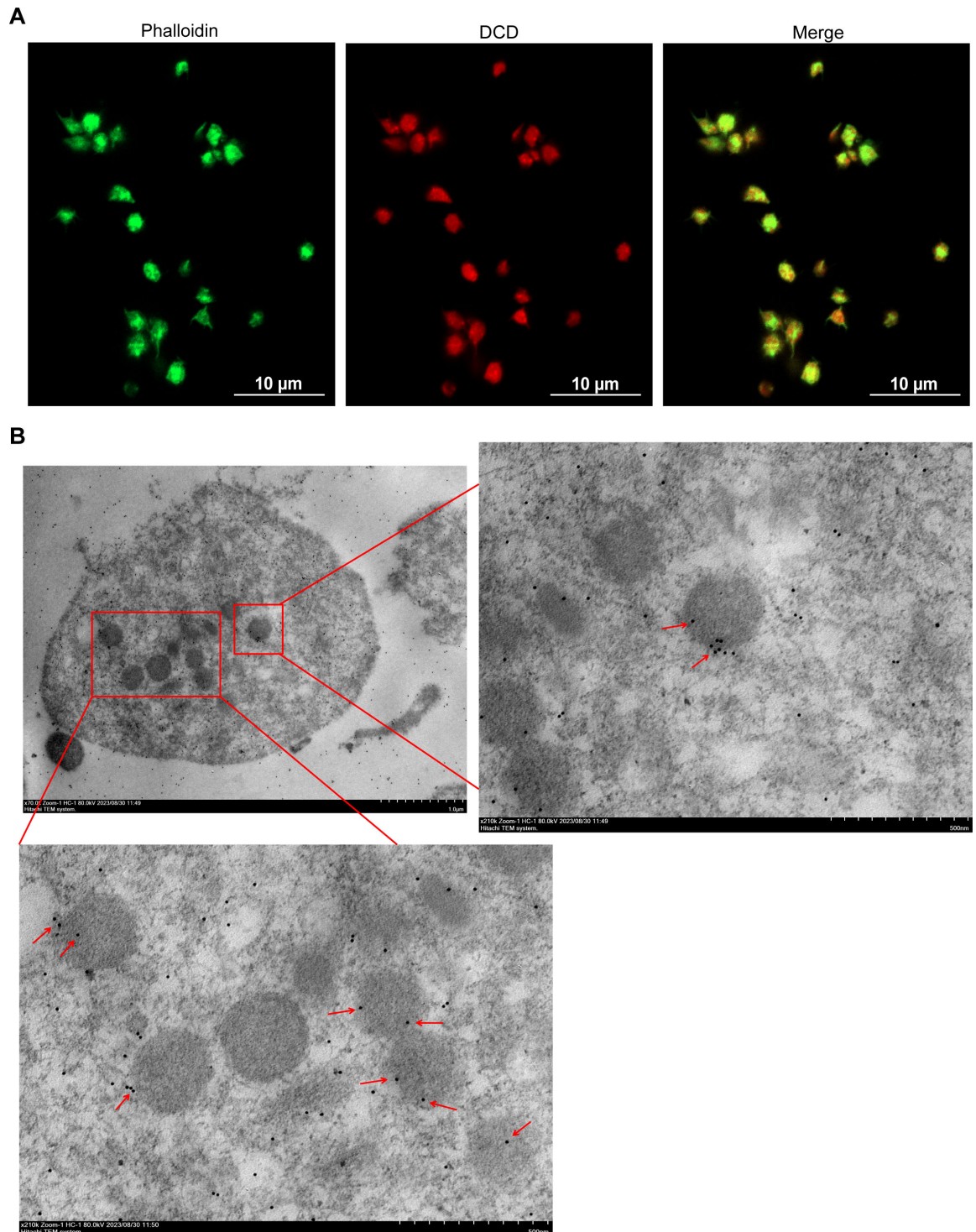

**FIG 4** DCD is present in the *α*-granules of platelets. (**A**) Observation of DCD in platelets by confocal microscopy. Phalloidin (green): CoraLite488 conjugated phalloidin, CL488-Phalloidin served as the counterstain, at dilution of 1:100; DCD (red): dermcidin. Scale bar = 10 µm. (**B**) Representative IEM image for the location of DCD in platelets. DCD is labeled with black gold beads (Red arrows indicate). Scale bar = 1.0 µm (left panel), 500 nm (right and below panels). The original magnification was 70,000× (left panel) and 210,000× (right and below panels). The panels in Fig. 4 are representative of three independent experiments.

Importantly, DCD promoted the inhibitory effect of platelets against MRSA after the addition of DCD-1L to the coculture system of platelets and MRSA.

Several studies have demonstrated antibacterial effects of platelets (49–51). When stimulated by *α*-toxin, platelets release *β*-defensin 1 (HBD-1). Platelet-derived HBD-1

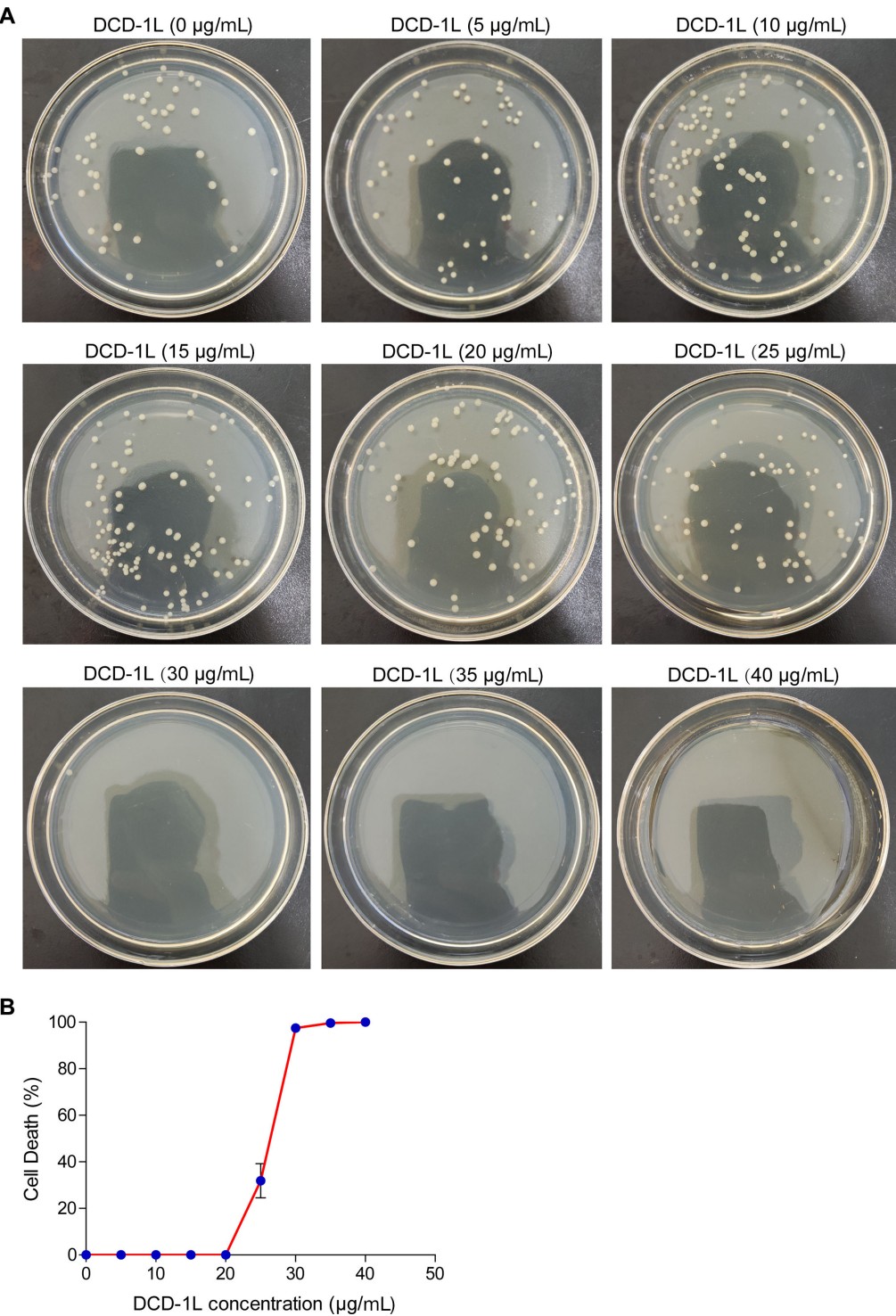

**FIG 5** Antimicrobial activity of DCD-1L against MRSA. MRSA ($10^6$ CFU/mL) was co-cultured with different concentrations of DCD-1L (0, 5, 10, 15, 20, 25, 30, 35, and 40 µg/mL) for 4 h, and then, the bacterial colony counts were counted by plates. (**A**) Bacteria count by plates. Images are representative of three independent experiments. (**B**) Bacterial mortality curve according to the results of (A). DCD-1L, Dermcidin-1L. All results have been tested at least three times. Data presented as mean ± SEM.

significantly inhibits *S. aureus* (52). Platelets directly regulate DNA damage and division in *S. aureus* (41). Our previous study showed that platelets could inhibit MRSA proliferation and that the inhibitory effect of platelet lysates on MRSA was consistent with that of platelets (40). This suggests that platelets perform antibacterial functions through the

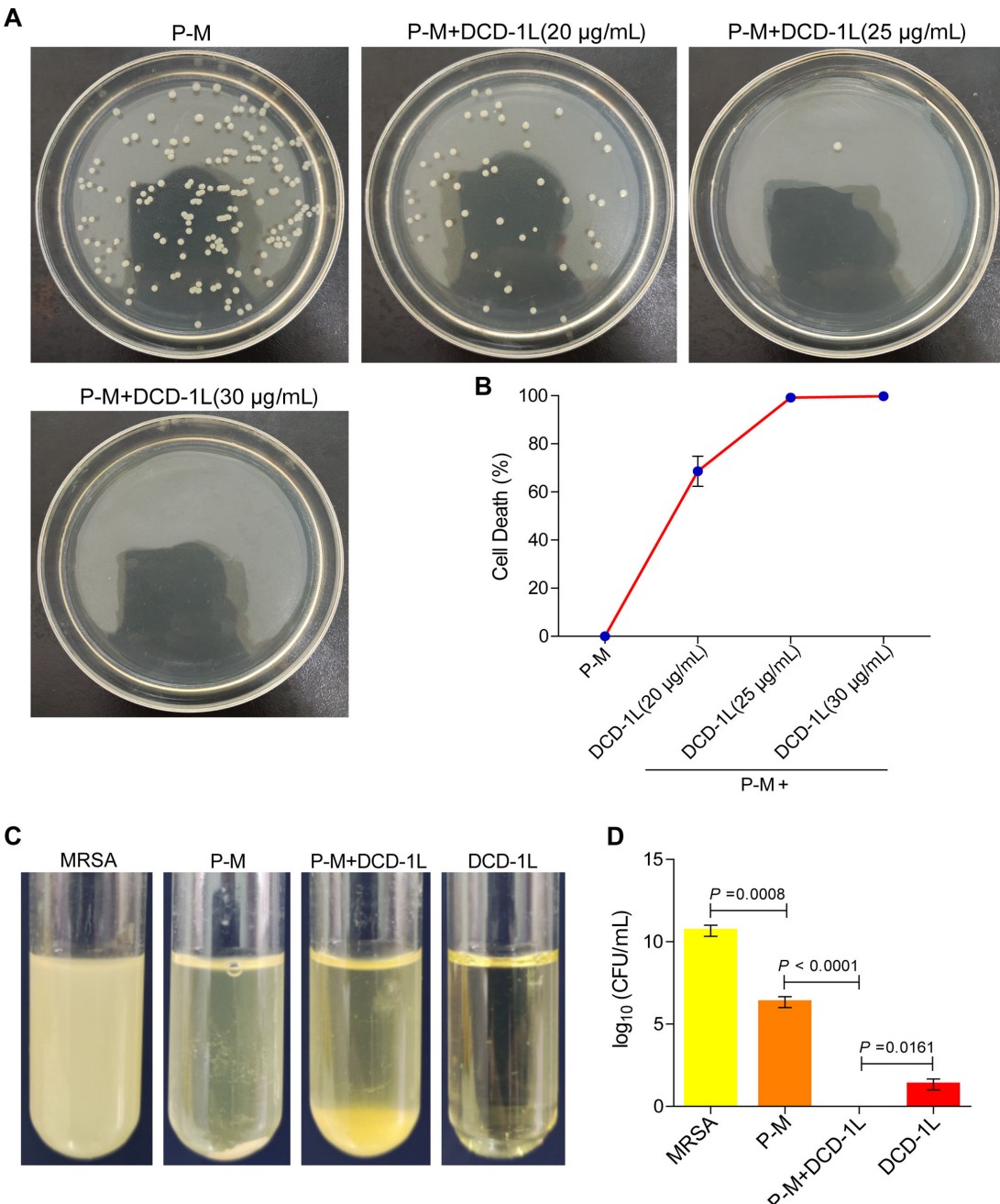

**FIG 6** DCD promoted the inhibitory effect of platelets against MRSA. (**A**) Bacterial colonies using plates. Adding different concentrations of DCD-1L to the bacterial suspension after platelet co-culture with MRSA for 10 h. Then, the bacterial colonies were counted on plates after DCD-1L incubated with the bacterial suspension for 4 h. Images are representative of three independent experiments. (**B**) Bacterial mortality curve according to the results of (A). P: platelet; P-M: platelets co-culture with MRSA; P-M + DCD-1L (20, 25, and 30 µg/mL): adding DCD-1L at concentrations of 20, 25, and 30 µg/mL, respectively, to the bacterial solution after platelets co-culture with MRSA for 10 h. (**C**) Bacterial fluid turbidity of each group. DCD-1L at a concentration of 30 µg/mL was added to the co-culture system of platelets and MRSA. After co-culture for 10 h, the turbidity of the bacterial suspension was compared, and the number of bacterial colonies was counted in each group. This figure is representative of three independent experiments. (**D**) Bacterial colony counts of each group in (C). MRSA: MRSA culture alone; P-M: platelets co-culture with MRSA; P-M +DCD-1L: adding DCD-1L (30 µg/mL) to the co-culture system of platelets and MRSA and then co-culture for 10 h. All results have been tested at least three times. Data presented as mean ± SEM. Student's *t* test for two-group comparisons.

antimicrobial proteins present in them. *α*-granules are key to the immune function of platelets. Platelet *α*-granules contain a variety of AMPs and kinocidins such as *β*-defensin

1, thymosin $\beta$4, thrombocidins 1 and 2, CXCL4, CXCL5, and CXCL7, all of which have significant antibacterial effects (53–55). In this study, we identified a previously unknown platelet antimicrobial peptide, DCD, in the $a$-granules. Large amounts of DCD were secreted from activated platelets after co-culturing with MRSA. DCD is an important natural antimicrobial peptide and was found in human sweat for the first time (46). DCD was initially reported to be constitutively expressed in human sweat glands (47). Recently, DCD has been shown to be expressed in breast cancer cells, hepatocytes, and human placental tissues (56–58).

The full-length DCD is composed of 110 amino acid residues. DCD is then hydrolyzed and processed by a protease *in vivo* to form C-terminal peptides of 48 amino acid residues (DCD-1L), 47 amino acid residues (DCD-1), and shorter fragments (42, 48). DCD-1L and DCD-1 show broad-spectrum antimicrobial activity against pathogens, including bacteria, fungi, and viruses, over a wide pH range and at high salt concentrations (59). The concentration of DCD in human sweat is 10 µg/mL (46). According to reports, the 90% inhibitory concentrations (IC$_{90}$) of DCD-1L against *S. aureus*, *S. epidermidis*, MRSA, and *E. coli* were 45, 10, 8, and 45 µg/mL, respectively, and DCD-1L could completely kill MRSA at the concentration of 10 µg/mL for 4 h (48). In this study, we found that DCD secretion increased significantly after platelets were cocultured with MRSA for 10 h, reaching a concentration of 3.58 µg/mL. This concentration was lower than the concentration of DCD-1L in sweat, and the concentration of DCD-1L found to completely inhibit MRSA growth in other studies. This may explain the partial antibacterial effects of platelets on MRSA (Fig. 6D). Additionally, we found that DCD-1L exerted antibacterial activities in a concentration-dependent manner. The IC$_{95}$ of DCD-1L against MRSA was 30 µg/mL, and DCD-1L showed 100% antibacterial activity on MRSA at a concentration of 35 µg/mL. The concentration of the antibacterial effect of DCD-1L on MRSA in our study was higher than that of DCD-1L against MRSA in previous studies, which may be because the DCD-1L we synthesized was not sufficiently pure, owing to differences in instruments and methods. In addition, we found that DCD-1L may promote platelet-induced inhibition of MRSA growth. Our findings provide credible evidence that platelet-derived DCD may contribute to inhibiting MRSA growth.

Although our study findings regarding novel platelet antimicrobial peptides for platelet-mediated inhibition of MRSA proliferation are noteworthy, there are still some limitations. We observed that DCD-1L inhibited MRSA growth *in vitro* and promoted platelet anti-MRSA effects. However, the *in vivo* antibacterial effect of DCD-1L on MRSA remains unknown and requires further verification. In addition, the antibacterial mechanisms of DCD remain unclear. Previous studies have found that DCD-1L binds to the bacterial surface instead of the cytoplasm and that the cell wall and membrane are not damaged (48). *S. aureus* was killed by DCD-derived peptides in a time-dependent manner, followed by membrane depolarization without pore formation (42, 60). Transcriptome analysis of the response of *S. epidermidis* to DCD-1L showed that DCD induced a general stress response (61). Our previous study showed that platelets induced oxidative stress and OH$^{\bullet}$ overproduction in MRSA, and excessive OH$^{\bullet}$ ultimately led to ALD in MRSA (40). Therefore, whether platelet-derived DCD also inhibits MRSA by inducing ALD of MRSA needs further investigation.

Our study is the first to identify a previously unrecognized platelet antimicrobial peptide, DCD, which impedes the proliferation of MRSA and promotes the antibacterial effect of platelets on MRSA. In the future, we will fully elucidate the mechanism by which platelet-derived DCD inhibits MRSA and further explore whether platelet-derived DCD has broad-spectrum antibacterial activity against other drug-resistant bacteria to develop new antimicrobial agents that can be used in clinical multidrug-resistant bacterial infections.

## ACKNOWLEDGMENTS

We thank the Department of Pathology, Fourth Military Medical University of China, for providing technical support for this study. This work was supported by the Natural

Science Basic Research Program of Shaanxi Province (grant no. 2024JC-YBQN-0890) and National Natural Science Foundation of China (grant no. 82170226 and 82370231).

E.L. and S.G. performed most of the experiments and analyzed the data; W.X. wrote the manuscript; W.X., L.Z., and J.X. discussed the ideas and revised the manuscript; W.W., J.X., N.A., and X.H. contributed to the assay, formal analysis, and discussed the results; Y.C., Q.A., and W.Y. designed this study, checked the manuscript, and interpreted the data.

## AUTHOR AFFILIATIONS

[1]Department of Transfusion Medicine, Xijing Hospital, Fourth Military Medical University, Xi 'an, Shaanxi, China
[2]Department of Geriatrics, Xi 'an North Hospital, Xi 'an, Shaanxi, China

## AUTHOR ORCIDs

Yaozhen Chen ⓘ http://orcid.org/0000-0003-2291-0622
Qunxing An ⓘ http://orcid.org/0009-0008-9609-3307
Wen Yin ⓘ http://orcid.org/0000-0002-7467-2286

## FUNDING

| Funder | Grant(s) | Author(s) |
|---|---|---|
| Natural Science Basic Research Program of Shaanxi Province | 2024JC-YBQN-0890 | Erxiong Liu |
| National Natural Science Foundation of China | 82170226 | Wen Yin |
| National Natural Science Foundation of China | 82370231 | Wen Yin |

## AUTHOR CONTRIBUTIONS

Erxiong Liu, Conceptualization, Data curation, Investigation, Methodology, Writing – original draft | Shunli Gu, Data curation, Investigation, Methodology, Visualization | Weizhen Xi, Formal analysis, Visualization, Writing – original draft, Writing – review and editing | Wenting Wang, Conceptualization, Methodology, Visualization | Jinmei Xu, Conceptualization, Methodology, Resources | Ning An, Resources, Software, Visualization | Lingling Zhang, Conceptualization, Formal analysis, Writing – review and editing | Jiajia Xin, Formal analysis, Software, Writing – review and editing | Xingbin Hu, Validation, Visualization, Writing – review and editing | Yaozhen Chen, Project administration, Supervision, Writing – review and editing | Qunxing An, Formal analysis, Project administration, Supervision | Wen Yin, Funding acquisition, Project administration, Resources, Supervision, Validation

## DATA AVAILABILITY

Data will be made available on request.

## ETHICS STATEMENTS

The studies involving humans were approved by the Medical Ethics Committee of Xijing Hospital, the Fourth Military Medical University. The studies were conducted in accordance with the local legislation and institutional requirements. The voluntary platelet donors provided their written informed consent to participate in this study.

## ADDITIONAL FILES

The following material is available online.

### Supplemental Material

**File S1 (Spectrum03273-24-S0001.docx).** Proteomic analysis techniques and methods.

**File S2 (Spectrum03273-24-S0002.xlsx).** The list of proteins identified by proteomics analysis.

**Legends (Spectrum03273-24-S0003.docx).** Legends for Files S1 and S2.

## Open Peer Review

**PEER REVIEW HISTORY (review-history.pdf).** An accounting of the reviewer comments and feedback.

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
