## [Reviewer comments · Microbiology Spectrum]

Microbiology Spectrum

Anti-bacterial Activity of Dermcidin in Human Platelets: Suppression of Methicillin-resistant *Staphylococcus aureus* Growth

Wen Yin, Erxiong Liu, Shunli Gu, Weizhen Xi, Wenting Wang, Jinmei Xu, Ning An, Lingling Zhang, Jiajia Xin, Xingbin Hu, Yaozhen Chen, and Qunxing An

Corresponding Author(s): Wen Yin, Fourth Military Medical University

Review Timeline:

Submission Date:	December 13, 2024
Editorial Decision:	March 11, 2025
Revision Received:	April 16, 2025
Accepted:	April 23, 2025

Editor: Nagendran Tharmalingam

Reviewer(s): Disclosure of reviewer identity is with reference to reviewer comments included in decision letter(s). The following individuals involved in review of your submission have agreed to reveal their identity: Diptaraj Chaudhari (Reviewer #2)

Transaction Report:

DOI: <https://doi.org/10.1128/spectrum.03273-24>

Re: Spectrum03273-24 (**Anti-bacterial Activity of Dermcidin in Human Platelets: Suppression of Methicillin-resistant *Staphylococcus aureus* Growth**)

Dear Prof. Wen Yin:

I have received the reviews of your manuscript and regret to inform you that we will not be able to publish it in Microbiology Spectrum. Your submission was read by reviewers with expertise in the area addressed in your study and it was the consensus view of these reviewers that your paper did not meet the standards necessary for publication.

I am sorry to convey a negative decision on this occasion, but I hope that the enclosed reviews are useful. Please note, rejections from Microbiology Spectrum are final and your manuscript will not be considered by other ASM journals. We wish you well in publishing this report in another journal and hope that you will consider Spectrum in the future.

Sincerely,
Nagendran Tharmalingam
Editor, Microbiology Spectrum

Reviewer #1 (Comments for the Author):

We thank the authors for submitting their work and my comments on the manuscript titled "Anti-bacterial Activity of Dermcidin in Human Platelets: Suppression of Methicillin-resistant *Staphylococcus aureus* Growth" are as follows:

1. The authors state that the replicates used for proteomic analysis are "good" however, based on Fig. 1 replicate three (P-3) appears significantly different from P-2 and -3. Overall, there are noticeable differences in the levels of expression of the same genes across replicates which may affect the data. Did the authors perform any corrections to maintain the validity of the data? Do the authors have more samples to include to strengthen the data?
2. Previous publications have shown the antimicrobial activity of dermcidin against MRSA (PMID: 1687074). Can the authors please elaborate on the contributions of this manuscript to the field?

Reviewer #2 (Comments for the Author):

The manuscript "Anti-bacterial Activity of Dermcidin in Human Platelets: Suppression of Methicillin-resistant *Staphylococcus aureus* Growth" is overall well written. The use of proteomic analysis, immunofluorescence, and antimicrobial assays, strengthens the findings of the study. Here are the comments for the further betterment of the manuscript. The manuscript can be accepted for publication after the successful revision.

Here are specific comments,

1. Abstract is well written and informative; the first statement could be more specific. It will be important to mention on the clinical significance of the MRSA.
2. Why was the proteomic analysis performed?
3. Line 63: the term "many drug-resistant bacteria" sounds vague and the statement can be modified as "The widespread use of antibiotics has led to the emergence of multidrug-resistant bacteria, which now pose a serious threat to public health"
4. The term "human washed platelets" is used inconsistently. Once clarified, it's better to stick with "platelets" throughout the section to avoid redundancy.
5. The methodology is redundant and can be concise and please mention the details only once in the section.
6. Line 254: was the total no of protein was same as that of the identified protein, or some unidentified proteins were also detected in the analysis. Please mention that too.
7. Wants the association between the pathways detected in the pathways and the antimicrobial function of the platelets.
8. The proteomic data should be submitted in the public domain data repositories.

The manuscript “Anti-bacterial Activity of Dermcidin in Human Platelets: Suppression of Methicillin-resistant Staphylococcus aureus Growth” is overall well written. The use of proteomic analysis, immunofluorescence, and antimicrobial assays, strengthens the findings of the study. Here are the comments for the further betterment of the manuscript. The manuscript can be accepted for publication after the successful revision.

Here are specific comments,

1. Abstract is well written and informative; the first statement could be more specific. It will be important to mention on the clinical significance of the MRSA.
2. Why was the proteomic analysis performed?
3. Line 63: the term “many drug-resistant bacteria” sounds vague and the statement can be modified as “The widespread use of antibiotics has led to the emergence of multidrug-resistant bacteria, which now pose a serious threat to public health”
4. The term "human washed platelets" is used inconsistently. Once clarified, it's better to stick with "platelets" throughout the section to avoid redundancy.
5. The methodology is redundant and can be concise and please mention the details only once in the section.
6. Line 254: was the total no of protein was same as that of the identified protein, or some unidentified proteins were also detected in the analysis. Please mention that too.
7. Wants the association between the pathways detected in the pathways and the antimicrobial function of the platelets.
8. The proteomic data should be submitted in the public domain data repositories.

Responses to Reviewer #1

Thank you very much for reviewer's comments and professional advice. The reviewer's comments concerning our manuscript, titled "Anti-bacterial Activity of Dermcidin in Human Platelets: Suppression of Methicillin-resistant *Staphylococcus aureus* Growth" (Spectrum03273-24), have helped us in improving the quality and academic rigor of this manuscript. We have gone through all the comments carefully and tried our best to incorporate the changes recommended. All the revised parts are marked in yellow in the Marked-Up Manuscript. Point-to-point responses to the reviewer's comments are given below:

1. *The authors state that the replicates used for proteomic analysis are "good" however, based on Fig. 1 replicate three (P-3) appears significantly different from P-2 and -3. Overall, there are noticeable differences in the levels of expression of the same genes across replicates which may affect the data. Did the authors perform any corrections to maintain the validity of the data? Do the authors have more samples to include to strengthen the data?*

Response: Thank you very much for your comments and professional advice. You have very accurately and promptly pointed out the mistakes and shortcomings of our manuscript. Indeed, our previous description in the manuscript was inappropriate and did not match the results shown in the figures. We promptly revised the description in the manuscript and corrected and reanalyzed the proteomics data to ensure the accuracy and validity of the data. We have added the revised description in page 10, lines 250-257 of the manuscript. And the revised result is shown in Figure 1. We earnestly thank you once again for your guidance and insightful suggestions.

2. *Previous publications have shown the antimicrobial activity of dermcidin against MRSA (PMID: 1687074). Can the authors please elaborate on the contributions of this manuscript to the field?*

Response: Thank you very much for your comments. You have very accurately and promptly pointed out the shortcomings of our study. Indeed, the previous study (PMID: 1687074) on the antimicrobial activity of dermcidin against MRSA bear similarities to the part results of our research. The key findings of our study comparing to previous work (PMID: 1687074) mainly consist of the following four points: 1. Dermcidin is a

previously-unrecognized platelet antimicrobial peptide, storing in the α -granules; 2. Platelets release large amounts of dermcidin after co-culture with MRSA; 3. Dermcidin inhibits MRSA growth in a concentration-dependent manner; 4. Dermcidin promotes the inhibition of MRSA growth by platelets. We previously confirmed that platelets inhibit MRSA by hydroxyl radical (OH[•])-mediated apoptosis-like cell death. And platelet lysates exert the same antibacterial effect as platelets^[1]. Therefore, it is possible that the anti-microbial proteins present in platelets play an important role in the anti-MRSA effects. Hence, which antibacterial protein derived from platelets contributes to the inhibition of MRSA growth require further investigation. In this study, we identified a previously-unknown platelet antimicrobial peptide, DCD, in the α -granules for the first time. We found that DCD display anti-MRSA activity in a concentration-dependent manner and promotes the antibacterial effect of platelets on MRSA. This enriches our understanding of platelet immune function and further improves the anti-MRSA mechanism of platelets. Additionally, our found in the present study provides theoretical and experimental basis for the research and development of new antibacterial agents and antibacterial therapy in the treatment of MRSA infection. We earnestly thank you once again for your guidance and insightful suggestions.

References

- [1] Liu E, Chen Y, Xu J, Gu S, An N, Xin J, Wang W, Liu Z, An Q, Yi J, Yin W. Platelets Inhibit Methicillin-Resistant Staphylococcus aureus by Inducing Hydroxyl Radical-Mediated Apoptosis-Like Cell Death. *Microbiol Spectr*. 2022 Aug 31;10(4):e0244121.

Responses to Reviewer #2

Thank you very much for reviewer's comments and professional advice. The reviewer's comments concerning our manuscript, titled "Anti-bacterial Activity of Dermcidin in Human Platelets: Suppression of Methicillin-resistant *Staphylococcus aureus* Growth" (Spectrum03273-24), have helped us in improving the quality and academic rigor of this manuscript. We have gone through all the comments carefully and tried our best to incorporate the changes recommended. All the revised parts are marked in yellow in the Marked-Up Manuscript. Point-to-point responses to the reviewer's comments are given below:

1. *Abstract is well written and informative; the first statement could be more specific. It will be important to mention on the clinical significance of the MRSA.*

Response: Thank you very much for your comments and professional advice. We have added the sentence as "As one of the most common superbugs and the pathogen with the highest global incidence of hospital-acquired infections, MRSA has developed resistance to multiple antibiotics, posing a serious threat to public health" on page 2, lines 27-30. We earnestly thank you once again for your guidance and insightful suggestions.

2. *Why was the proteomic analysis performed?*

Response: Thank you very much for your comments. We previously confirmed that platelets inhibit MRSA by inducing hydroxyl radical (OH[•])-mediated apoptosis-like cell death (ALD). And platelet lysates exert the same antibacterial effect as platelets. Therefore, it is possible that the anti-microbial proteins present in platelets play an important role in the anti-MRSA effects. Hence, in order to clarify which antibacterial protein derived from platelets contributes to the inhibition of MRSA growth, we conducted proteomic analysis and carried out a preliminary screening of proteins derived from platelets. Proteomics analysis laid an important foundation for the discovery of dermcidin (DCD), a previously-unrecognized antimicrobial peptide of platelets, in this study. The relevant descriptions have been added on page 10, lines 243-248. We earnestly thank you once again for your guidance.

3. *Line 63: the term "many drug-resistant bacteria" sounds vague and the statement can be modified as "The widespread use of antibiotics has led to the emergence of multidrug-resistant bacteria, which now pose a serious threat to public health".*

Response: Thank you very much for your professional advice. The term "With the widespread use of antibiotics, many drug-resistant bacteria, particularly multidrug-resistant bacteria, have emerged and pose a serious threat to public health" have been modified as "The widespread use of antibiotics has led to the emergence of multidrug-resistant bacteria, which now pose a serious threat to public health". The revised statements have been added to page 3, lines 66-67.

4. The term "human washed platelets" is used inconsistently. Once clarified, it's better to stick with "platelets" throughout the section to avoid redundancy.

Response: Thank you very much for your guidance and insightful suggestions. We have revised the relevant descriptions in 2.2, 2.3, 2.4, 2.6 and 2.7 of the method section, uniformly using the term "Platelets".

5. The methodology is redundant and can be concise and please mention the details only once in the section.

Response: Thank you very much for your comments and professional advice. We have simplified the methods and removed the repetitive details in each method. The revised method descriptions can be found in the methods section of the manuscript and marked up by highlighting.

6. Line 254: was the total no of protein was same as that of the identified protein, or some unidentified proteins were also detected in the analysis. Please mention that too.

Response: Thank you very much for your comments and professional advice. The total number of proteins is different from the number of identified proteins. During the analysis, some unidentified proteins were also detected. The accurate expression is as follows: "In total, 2358 human platelet proteins were detected. Among them, 2304 proteins with unique peptides or polypeptide segments were identified, while 58 proteins were not identified." The revised statements have been added on page 10, lines 250-252. We earnestly thank you once again for your guidance and insightful suggestions.

7. Wants the association between the pathways detected in the pathways and the antimicrobial function of the platelets.

Response: Thank you very much for your comments. Platelets are multifunctional granulocytes with extensive sensors to sense microbial pathogens. Stimuli related to infection will lead to rapid activation of platelets^[1]. The α granules of activated platelets will release a large amount of antibacterial proteins to exert antibacterial effects^[2,3,4]. Therefore, platelet activation is a prerequisite for platelets to exert antibacterial functions. In this study, through proteomics analysis, we discovered the platelet activation pathway. And activated platelets release large amounts of dermcidin (DCD), a previously unknown antimicrobial peptide exists in the α -granules of platelets, after co-culture with MRSA. Therefore, the platelet activation pathway provided important enlightenment for the subsequent research on the antibacterial function of platelets. We earnestly thank you once again for your guidance and insightful suggestions.

References

- [1] Yeaman MR: Platelets: at the nexus of antimicrobial defence[J]. NATURE REVIEWS MICROBIOLOGY 2014, 12:426-437.
- [2] Price Blair PDaRF: Platelet α -granules: Basic biology and clinical correlates[J]. Blood Rev 2009, 23(4):177-189.
- [3] Yeaman MR: Bacterial-platelet interactions: virulence meets host defense[J]. Future Microbiol 2010, 5(3):471-506.
- [4] Yeaman MR, Yount NY: Unifying themes in host defence effector polypeptides[J]. Nat Rev Microbiol 2007, 5(9):727-740.

8. The proteomic data should be submitted in the public domain data repositories.

Response: Thank you very much for your professional advice. We will submit the proteomics data to the public domain data repositories as per the requirements of this journal. We earnestly thank you once again for your guidance and insightful suggestions.

Re: Spectrum03273-24R1-A (**Anti-bacterial Activity of Dermcidin in Human Platelets: Suppression of Methicillin-resistant *Staphylococcus aureus* Growth**)

Dear Prof. Wen Yin:

Your manuscript has been accepted, and I am forwarding it to the ASM production staff for publication. Your paper will first be checked to make sure all elements meet the technical requirements. ASM staff will contact you if anything needs to be revised before copyediting and production can begin. Otherwise, you will be notified when your proofs are ready to be viewed.

Sincerely,
Nagendran Tharmalingam
Editor
Microbiology Spectrum